# Assessment of physical status and analysis of lipidomic and metabolomic alterations in patients with Post-COVID-19 condition

Raúl Pavón[1,2], Sandra Parra [1,2]*, Francisco Javier Rubio[3,2], Mireia Feliu[1,2‡],
Marta Ríos[1,2‡], Simona Iftimie[1,2‡], Conxita Rovira[4‡], Nuria Amigó[5,6,7], Lydia Cabau[5,6],
Neus Martínez-Micaelo[5,6], Antoni Castro[1,2]

1 Internal Medicine Department, "Sant Joan" University Hospital, Reus, Spain, 2 Grup Autoimmunitat, infecció i trombosi (GRAIIT), Institut Investigació Sanitària Pere Virgili (IISPV), Universitat Rovira i Virgili (URV), Reus, Spain, 3 Sports Medicine Department, "Sant Joan" University Hospital, Reus, Spain, 4 Intensive Care Unit, "Sant Joan" University Hospital, Reus, Spain, 5 Biosfer Teslab, Reus, Spain, 6 Department of Basic Medical Sciences, Universitat Rovira I Virgili (URV), Institut Investigació Sanitària Pere Virgili (IISPV), Tarragona, Spain, 7 Centre for Biomedical Research Network on Diabetes and Associated Metabolic Diseases (CIBERDEM), ISCIII, Madrid, Spain

☉ These authors contributed equally to this work.
‡ MF, MR, SI and CR also contributed equally to this work.
* sandra.parra@urv.cat

## Abstract

The development and persistence of symptoms following SARS-CoV-2 infection, known as Post-COVID-19 Condition (PCC) or "long COVID," represents a global health challenge. In this prospective cross-sectional study, we conducted a detailed assessment of the physical condition of 46 patients using handgrip dynamometry, ergoespirometry, and the 6-minute walk test (6MWT). The results revealed a loss of muscle strength and poor exercise tolerance primarily due to peripheral muscle involvement. To complement and better understand these findings, we compared the blood metabolome and lipidome of 13 patients with PCC, 13 patients with acute COVID-19 infection, and 13 healthy controls using magnetic resonance spectroscopy (1H-NMR). PCC patients showed lower levels of HDL-cholesterol, as well as medium and dense HDL particles, which could contribute to a pro-atherogenic and pro-inflammatory state. Although no significant differences were observed in glycoproteins, we found decreased glucose and increased lactate levels, supporting the hypothesis of mitochondrial dysfunction in PCC patients. Additionally, elevated glycine and reduced glutamate levels may be related to the neurological symptoms associated with the condition. We also observed increased levels of glutamine, leucine, and isoleucine, indicating protein hypercatabolism and metabolic stress. These findings suggest that alterations in the metabolome and lipidome of PCC patients may be contributing to the persistence of their symptoms.

**Data availability statement:** All relevant data are available in a public repository at the following link: https://doi.org/10.34810/data2900.

**Funding:** The author(s) received no specific funding for this work.

**Competing interests:** The authors have declared that no competing interests exist.

## Introduction

The Coronavirus Disease 2019 (COVID-19), caused by the SARS-CoV-2 coronavirus infection, has had a major global impact both socioeconomically and health levels [1]. The acute phase with respiratory involvement includes the potential development of severe respiratory failure due to adult respiratory distress syndrome (ARDS) [2]. However, after overcoming the acute phase of the infection, the identification of the so-called Post-COVID-19 Condition (PCC) highlights the long-term challenge posed by this infectious disease [3].

Post-COVID-19 Condition is defined as the persistence of symptoms or the development of sequelae after 12 weeks from the onset of the acute clinical episode [4–6], demonstrated by analytical tests, such as antigen tests or polymerase chain reaction (PCR). These symptoms must persist for at least 8 weeks and cannot be justified by other diagnoses [7]. PCC encompasses a wide clinical spectrum, which includes not only respiratory symptoms but also gastrointestinal, cardiovascular, neuropsychiatric, or musculoskeletal manifestations [8,9]. Among the most frequently reported symptoms are fatigue, dyspnea, neurocognitive impairments, and arthralgias or myalgias [10].

The diverse clinical presentations and their impact on patients live underline the need for multidisciplinary and personalized management to optimize diagnosis and improve the quality of life for these patients.

Although the proper management of PCC patients remains without clear standardization or guidelines, the utility of physical rehabilitation programs is becoming increasingly evident, aiming to improve symptoms and the quality of life of these patients. The benefits of pulmonary rehabilitation appear well-established, with the goal of enhancing the respiratory capacity of affected individuals. Similarly, guidelines and protocols for mild regular aerobic exercise also show promising results in addressing sarcopenia stemming from this infection, especially during the acute phase of the illness [11–14].

Following this approach and considering the benefits of these rehabilitation programs, understanding the state of physical fitness as well as the metabolic alterations associated with this condition becomes particularly relevant as to way to provide insights on the pathophysiology of the disease, help identify new biomarkers for diagnosis, and enable a more appropriate therapeutic management.

The metabolome is the collection of metabolites present in an organism, tissues, or cells at a given time. Metabolite profiling has been used to better understand the pathophysiological processes of various diseases and to identify new biomarkers for their diagnosis and treatment [15].

Previous studies have shown how different viral infections can alter our metabolic state by studying changes in the metabolome using nuclear magnetic resonance (1H-NMR) spectroscopy. This technique detects alterations in lipid, carbohydrate and low-molecular-weight metabolites metabolism. It can even identify which organs and tissues are affected by infections [16, 17]. The detection of glycoprotein peaks via NMR has emerged as a highly promising inflammatory marker in these chronic conditions [18]. Similarly, this technique has been employed to study metabolic changes during the acute phase of SARS-CoV-2 infection, aiding in the identification of diagnostic and severity biomarkers [19].

Moreover, significant metabolic changes have already been persistently identified after the acute phase of COVID-19. For instance, persistent alterations in glucose metabolism have been observed, including a decrease in its utilization as an energy source. This leads to an increase in ketone bodies and, as a result, a condition of persistent metabolic stress [20].

For these reasons, we propose a study of the lipidome, metabolome, and glycoprotein peaks using NMR in patients with PCC compared to hospitalized patients with acute SARS-CoV-2 infection and healthy controls.

## Materials and methods

### Population

This is a cross-sectional observational study that included an initial group of 46 patients with PCC, who underwent a physical condition assessment at the Sports Medicine Unit of the "Sant Joan" University Hospital in Reus (Tarragona, Spain). From this group of patients, blood samples were obtained from a total of 13 patients for metabolomic and lipidomic analysis using NMR.

The diagnosis and recruitment of patients with PCC were carried out between July 2021, and June 2022. Blood samples were obtained between July 2021 and July 2022, and were accessed and analyzed between July 2022 and February 2023. The physical condition assessment was carried out shortly after the diagnosis and was extended throughout 2023, until December 2023.

In total, blood samples were collected from three groups of patients (n = 39), matched by age, gender, and body mass index (BMI, measured in kg/m$^2$). These groups included 13 patients during the acute phase of SARS-CoV-2 infection, 13 healthy control individuals, and 13 patients diagnosed with PCC, who also underwent a physical condition assessment.

For the acute infection group, SARS-CoV-2 infection was confirmed using the VIASURE SARS-CoV-2 Real-Time PCR Detection Kit (CerTest Biotec, Zaragoza, Spain). All of them required hospitalization in the Intensive Care Unit (ICU) of the "Sant Joan" University Hospital in Reus (Tarragona, Spain) and oxygen therapy (determined by an oxygen saturation level measured by pulse oximetry of less than 95% or arterial oxygen pressure below 80 mmHg). Patients with PCC who participated in the study were recruited from the outpatient clinic dedicated to Post-COVID-19 Condition follow-up in the Internal Medicine Department of the hospital. These PCC patients met the diagnostic criteria proposed by the World Health Organization (WHO).

Clinical, analytical, and anthropometric variables, as well as a history of cardiovascular risk factors and comorbidities, were collected from the study population.

All patients were over 18 years old and freely agreed to participate in the study by signing an informed consent form. Verbal consent was obtained from those patients who only participated in the physical condition assessment, without obtaining blood samples. The study complies with the regulations of the Declaration of Helsinki and the Belmont Report for clinical studies and was previously approved by the Ethics Committee of the Pere Virgili Health Research Institute (IISPV-HUSJR, ref. 088/2021).

### Analysis of physical condition in PCC patients

The 46 patients diagnosed with PCC in this study underwent a detailed physical condition assessment with the participation of the Sports Medicine Department at the "Sant Joan" University Hospital in Reus. All participants underwent dynamometry using a KERN MAP 80K1 hand dynamometer (Kern & Sohn, Albstadt, Germany), a 6-minute walk test, and ergoespirometry

Ergoespirometry was conducted using an ERGOCARD-MEDISOFT ergospirometer and a Corival model cycle ergometer (Ergometrix, Esplugues de Llobregat, Spain). It was not possible to perform ergoespirometry on a larger sample of the PCC group due to their poor physical condition, which did not allow for the achievement of minimum evaluable values.

## Lipidome and Metabolome analysis by 1H-NMR spectroscopy

For the three groups of patients (COVID, post-COVID, and controls) from whom plasma samples were obtained, nuclear magnetic resonance spectroscopy (1H-NMR) analysis was performed. This included profiling of lipoproteins, glycoproteins, lipids, and low-molecular-weight metabolites (LMWM).

Frozen samples were shipped on dry ice to Biosfer Teslab (Reus, Spain) for 1H-NMR analysis. Prior to NMR analysis, serum samples were thawed overnight and homogenized. High-resolution 1H-NMR spectra were acquired on a Bruker Avance III 600 MHz spectrometer (Bruker BioSciences Española S.A., Rivas Vaciamadrid, Madrid, Spain) operating at a proton frequency of 600.20 MHz (14.1 T) and 310 K. We used the LED (ledbpg-p2s1d) pulse sequence -a double stimulated echo with bipolar gradients and longitudinal eddy current delay- together with 1D NOESY experiments, and Carr-Purcell-Meiboom-Gill (CPMG) were used to characterize small molecules such as amino acids and sugars. The relaxation delay was 2 s, and 64k complex points were collected for each scan.

Control and study samples consisted of 200 μL serum, 300 μL of 50 mM phosphate buffer (pH 7.4), and 50 μL $D_2O$, transferred into 5 mm NMR tubes. Samples were maintained below 8 ºC until loading into the spectrometer. After insertion into the probe, a minimum of 5 minutes was allowed for temperature equilibration.

All acquisition steps (tuning, matching, pulse calibration, receiver gain, number of scans) were performed through automatic Bruker routines. Quantification relied on a reference quality control (QC) sample and corrected during acquisition according to pulse length, number of scans, and receiver gain.

To ensure year-round quality control, we use an internal pooled serum control prepared in ~100 aliquots (200 μL each) and stored at −80 ºC. One aliquot is analyzed at the beginning of each analytical day. We guarantee <1% variability in the region 1.9–0.4 ppm. Spectra from different days are overlaid in TopSpin 3.2 (Multiple Display) to ensure no intensity shifts. (Examples are provided in S1 Fig.).

Automated spectral processing is integrated into the Liposcale® software, ensuring identical conditions for all samples. The workflow includes: Phase correction, Baseline correction and Spectral referencing and alignment. Referencing is performed using the glycoprotein region (2.15–1.90 ppm), aligning the main peak at 2.034 ppm. This guarantees consistent spectral positioning across samples. (Examples are provided in S2 Fig.).

## Lipoprotein analysis by 1H-NMR spectroscopy (extended lipoprotein profile)

The lipoprotein profile was analyzed using the NMR-based Liposcale® assay. The Liposcale® test has been continuously optimized since 2012, including sample handling conditions, probe temperature management, signal consistency, and software development. It has obtained CE marking, ISO 13485 certification, and approval by the Spanish Agency for Medicines and Health Products. It is currently implemented in both public and private healthcare settings in Spain.

Lipid concentrations (i.e., triglycerides and cholesterol), the size and particle number of the four major lipoprotein classes (intermediate-density lipoprotein, IDL; very-low-density lipoprotein, VLDL; low-density lipoprotein, LDL; and high-density lipoprotein, HDL), as well as nine lipoprotein subclasses (large, intermediate, and small VLDL, LDL, and HDL) were determined as previously reported [21]. The particle concentration of each subclass was obtained by dividing the lipid volume by the particle volume of the corresponding class, using established conversion factors to transform concentration units into volume units. After preprocessing, signal deconvolution of the methyl region (~0.8 ppm) was performed, where each fitted function represents a specific lipoprotein class (VLDL, LDL, HDL) and subparticle size. Lipoprotein size ranges were defined as follows: VLDL: 38.6–81.9 nm, LDL: 14.7–26.6 nm, and HDL: 6.0–10.9 nm. Weighted mean VLDL, LDL, and HDL particle sizes were then calculated by summing the known diameter of each subclass multiplied by its relative percentage of particle number, and averaging areas across fitted functions to obtain the final mean particle size for each lipoprotein class.

## Glycoprotein profiling using 1H-NMR spectroscopy

The glycoprotein profile was determined by analyzing the region of the 1H-NMR spectrum where the glycoproteins resonate (2.15–1.90 ppm) using several analytical functions according to a previously published procedure. The H/W ratios of GlycA and GlycB were also reported as a parameter related to the aggregation state of the sugar-protein linkages [18]. Height was calculated as the difference from baseline to maximum of the corresponding NMR peaks, and width value corresponds to the peak width at half height.

## Analysis of low molecular weight metabolites using 1H-NMR spectroscopy

Beyond the lipoprotein and glycoprotein profiling, for the analysis of LMWM, we acquired CPMG spectra under the same experimental conditions described above (600 MHz, 310 K, identical sample preparation and equilibration times). The same pooled human serum QC sample used daily for lipoprotein profiling was also analysed using the CPMG experiment to monitor the stability of small-molecule resonances and to ensure day-to-day reproducibility. Metabolite assignment was performed using chemical-shift libraries and characteristic multiplicity patterns [22,23], and quantification was carried out on processed CPMG spectra after standard phase correction, baseline correction, and referencing at 2.034 ppm. This unified QC strategy ensures full consistency and comparability between lipoprotein and low-molecular-weight metabolite measurements. Day-to-day spectral variability remains below 5%, with weekly QC measurements and biannual instrument calibration to detect potential acquisition bias.

## Statistical analysis

Quantitative results were expressed as mean and standard deviation for continue variables with normal distribution or with the median and 25–75 interquartile range otherwise, while qualitative or dichotomous variables were presented as percentages. To compare proportions and assess relationships between qualitative variables, we used the Chi-square test ($\chi^2$) or Fisher's exact test, depending on sample size and expected frequencies. For comparisons of quantitative variables between groups, ANOVA or Kruskal–Wallis tests were applied, and Wilcoxon tests were used for pairwise non-parametric comparisons. In addition, Spearman's Rho correlation coefficients were calculated to estimate associations between continuous variables.

All statistical analyses were performed using R statistical software (version 4.3.3, R Foundation for Statistical Computing, Vienna, Austria).

# Results

### General characteristics of the study population

Among the 46 patients that performed the sport medicine evaluation, with a predominantly female population (60.87%), the average age was 48, and the mean body mass index was 27.93 kg/m². Regarding comorbidities and prior medical history, the sample showed a 32.61% rate of obesity. Hypertension was present in 21.74% of the individuals, and 23.91% had dyslipidemia or were tobacco users. A prevalence of 8.70% of prior cardiovascular disease was identified within the sample, along with 6.52% with oncological disease, 4.35% with respiratory disease, and 26.09% with psychiatric disorders. During the acute phase of COVID-19, 28.26% of the sample required hospitalization, and 17.39% needed intensive care unit admission.

If we review the persistent symptoms related to PCC that they suffer, 82.10% presented respiratory symptoms (including dyspnea, cough, or sputum production), 28.26% presented digestive symptoms (including abdominal pain or changes in bowel habits), 50% presented neurological symptoms (including headaches, concentration issues, or brain fog), and 69.57% presented systemic symptoms (including fatigue, arthralgia, or myalgia).

Regarding the 6-minute walk test, a mean of 437.04m with a standard deviation of 113.47 has been obtained. 16 did not reach the threshold of 400 meters walked (34.78% of the sample). When adjusting results for sex, age, weight, and height using standardized formulas [24,25], we found an average distance of 70% compared to the ideal target distance. A desaturation below 95%, measured by fingertip pulse oximetry, was observed in two of the included patients.

Dynamometry results were also obtained, with measurements taken on the dominant hand. A mean of 22.41 Kg with a standard deviation of 7.10 has been obtained. According to reference tables of average values in a healthy population adjusted for age and sex [26–28], only one patient reached the expected average strength values (2.17% of the sample).

Regarding ergoespirometric parameters, the average maximum heart rate was 133.08 beats per minute, corresponding to 76.10% of the theoretical maximum. An average workload of 86.02 watts was achieved, representing only 55.37% of the theoretical maximum. Finally, the mean maximum oxygen consumption (VO2max) was 17.30 ml/kg/min, equivalent to reaching 61.28% of the theoretical expected value, representing an average VO2max decrease of 38.72%. These results demonstrate poor physical condition in the studied population. Table 1.

### Metabolome analysis by 1H-NMR in three subgroups of PCC patients, compared to control healthy group and patients with acute COVID-19

A total of 39 patients were included, divided into three groups matched by age, sex, BMI, and cardiovascular risk factors. These groups consisted of an initial group of healthy controls (n = 13), a group of patients with active SARS-CoV-2 infection (n = 13), and a group of patients diagnosed with PCC (n = 13).

Reviewing the cardiovascular risk factors present in the study population, no significant differences were found in the prevalence of arterial hypertension (HBP), type 2 diabetes mellitus (T2D), or obesity among the three study groups. Table 2.

Regarding patients affected by PCC, among the persistent related symptoms, 11 of them (84.60%) presented respiratory symptoms (including dyspnea, cough, or sputum production), 3 of them (23.07%) presented digestive symptoms (including abdominal pain or changes in bowel habits), 5 of them (38.46%) presented neurological symptoms (including headaches, concentration issues, or brain fog), and 8 of them (61.53%) presented systemic symptoms (including fatigue, arthralgia, or myalgia).

### Lipoprotein profile analysis performed by 1H-NMR

First, we observed a higher presence of pro-atherogenic lipoproteins in the acute COVID-19 group compared to the control group and the Post-COVID-19 group. This was evident in VLDL-cholesterol [23.7 (18.7–31.5) compared to 9.92 (8.61–14.0) and 12.0 (10.5–14.2) mg/dL, respectively, p < 0.001]; VLDL-triglycerides [78.6 (70.6–121) compared to 47.7 (35.0–70.1) and 59.8 (48.2–69.1) mg/dL, p = 0.003]; IDL-cholesterol [19.9 (14.0–26.4) compared to 8.08 (6.14–10.1) and 8.33 (6.88–9.70) mg/dL, p < 0.001]; and IDL-triglycerides [18.1 (14.1–23.9) compared to 9.27 (7.93–10.1) and 9.84 (7.43–10.8) mg/dL, p < 0.001]. Similarly, we detected a greater presence of VLDL particles during acute infection compared to the control group and the PCC group [59.5 (53.7–90.3) compared to 33.1 (25.8–46.9) and 42.2 (34.6–48.2) nM, p = 0.002]. These particles tend to be medium and small in size.

Total triglycerides followed a similar pattern [143 (122−211) compared to 79.6 (65.7−103) and 89.2 (83.0−109) mg/dL, p < 0.001], indicating a trend toward an atherogenic dyslipidemia in the acute infection group. During the acute phase, we also observed a tendency for LDL particles to be richer in triglycerides compared to cholesterol, compared to the control group and the PCC group: LDL-TG/LDL-C [0.25 (0.18–0.29) compared to 0.09 (0.08–0.10) and 0.09 (0.08–0.11), p < 0.001].

Regarding high-density lipoproteins (HDL), during the acute phase, HDL particles, like LDL particles, tended to be rich in triglycerides. However, we observed that this decrease in HDL-cholesterol persisted in PCC patients [42.2 (33.7–43.4) COVID group, 54.0 (50.4–56.5) Post-COVID group, 68.7 (66.3–71.4) mg/dL control group, p < 0.001]. In PCC patients,

**Table 1. Population characteristics and results of the medical-sports analysis conducted on patients with post-COVID-19 condition.**

| GENERAL CHARACTERISTICS OF THE PCC POPULATION (n=46) | |
|---|---|
| Age, (years) | 48 (12.4) |
| Women (%) | 60.87 |
| BMI (kg/m2) | 27.93 (5.37) |
| Obesity (%) | 32.61 |
| High blood pressure (%) | 21.74 |
| Dyslipidemia (%) | 23.91 |
| T2D (%) | 8.70 |
| Smokers (%) | 23.91 |
| Cardiovascular disease (%) | 8.70 |
| Oncological disease (%) | 6.52 |
| Respiratory disease (%) | 4.35 |
| Psychiatric disease (%) | 26.09 |
| Hospital admission (%) | 28.26 |
| ICU admission (%) | 17.39 |
| **PREVALENCE OF PERSISTENT SYMPTOMS (n=46)** | |
| Respiratory symptoms (%) | 82.10 |
| Digestive symptoms (%) | 28.26 |
| Neurological symptoms (%) | 50.00 |
| Systemic symptoms (%) | 69.57 |
| **6MWT (n=46)** | |
| Distance (m) | 437.04 (113.47) |
| Heart rate (bpm) | 112.15 (14.15) |
| SpO2 (%) | 97.50 (1.44) |
| **DINAMOMETRY (n=46)** | |
| Best try (Kg) | 22.41 (7.10) |
| **ERGOESPIROMETRY (n=46)** | |
| Maximal heart rate (bpm) | 133.09 (24.47) |
| Maximal heart rate (%) | 76.11 (11.55) |
| Load (Watts) | 86.02 (49.05) |
| Load (%) | 55.37 (27.32) |
| Maximal O2 consumption – VO2max (ml/Kg/min) | 17.30 (9.12) |
| Maximal O2 consumption – VO2max (%) | 61.28 (23.60) |

Continue variables are represented as mean and standard deviation in parenthesis. BMI, body mass index; T2D, type 2 diabetes; ICU, intensive care unit.

we also found a reduction in medium and small HDL particles. The fact that HDL particles in the PCC group tended to be larger in size could also promote a pro-atherogenic environment. Fig 1. S1 Table.

## Analysis of the Glycoprotein profile by 1H-NMR

In the present study, we observed significant alterations in the glycoprotein profile obtained by 1H-NMR in patients with COVID-19 infection. The serum glycoproteins evaluated through this technique correspond predominantly to acute-phase reactants, whose concentration and glycosylation patterns are modified in inflammatory contexts. Notably, the signal peak corresponding to Glyc-A was markedly elevated [982 (868–1183) COVID group, 627 (612–645) Post-COVID group, 592

**Table 2. General characteristics of the control group, active infection group (COVID), and post-COVID-19 condition group (post-COVID), that underwent metabolome study by 1H-NMR.**

| | Control n=13 | COVID n=13 | post-COVID n=13 | p-value |
|---|---|---|---|---|
| **Gender** | | | | 0.233 |
| Female, % | 12 (92.3%) | 8 (61.5%) | 10 (76.9%) | |
| Male; % | 1 (7.69%) | 5 (38.5%) | 3 (23.1%) | |
| **Age, years** | 54.0 [52.0-57.0] | 54.0 [49.0-59.0] | 51.0 [45.0-55.0] | 0.374 |
| **BMI, kg/m2** | 28.6 [25.0-29.0] | 26.4 [25.8-37.6] | 27.6 [24.0-30.7] | 0.798 |
| **HBP, %** | 0 (0.00%) | 2 (15.4%) | 1 (7.69%) | 0.760 |
| **T2D, %** | 0 (0.00%) | 3 (23.1%) | 1 (7.69%) | 0.297 |
| **Obesity, %** | 10 (76.9%) | 6 (46.2%) | 5 (38.5%) | 0.115 |
| **ICU admission; %** | 0 (0.00%) | 13 (100%) | 0 (0.00%) | <0.001 |

Continue variables are presented as median and 25–75 interquartile range. BMI, body mass index; HBP, high blood pressure; T2D, type 2 diabetes; ICU, intensive care unit.

(526–672) µM control group, p<0.001], a finding previously associated with systemic inflammatory states as well as atherosclerosis [29,30]. However, we did not observe these differences in Post-COVID patients, except for Glyc-F, where the results were not statistically significant [205 (182–257) COVID group, 208 (196–231) Post-COVID group, 188 (167–196) µM control group, p=0.300]. Results are shown in Fig 2. S2 Table.

### Lipid Profile analysis performed by 1H-NMR

A significant decrease in sphingomyelin levels was observed in PCC patients compared to the control group and acute infection patients [0.81 (0.78–0.89) compared to 1.00 (0.94–1.02) and 0.92 (0.80–1.00) mM, p=0.023]. Results are shown in Fig 3. S3 Table.

Continuing with lipoproteins and their pro-atherogenic and pro-inflammatory roles, we found an increase in triglycerides in the COVID group, which was not sustained in PCC patients [1.54 (1.20–2.64) COVID group, 0.80 (0.48–0.97) Post-COVID group, 1.23 (0.79–1.58) mM control group, p=0.022]. However, regarding omega-3 fatty acids, known for their protective effects against atherosclerosis, no statistically significant differences were found among the three groups [31].

### Analysis of low molecular weight metabolites performed by 1H-NMR

Significant differences were observed in the profile of low molecular weight metabolites. PCC patients exhibited a significant increase in lactate levels compared to the control group. This increase, also present during the acute infection phase, did not normalize to control values in PCC patients [1007 (881–1222) COVID group, 390 (313–442) Post-COVID group, 237 (151–350) µM control group, p<0.001]. Combined with decreased glucose levels in the PCC group [4339 (3950–6140) COVID group, 3054 (2732–3588) Post-COVID group, 4205 (3733–4515) µM control group, p=0.005], this points to oxidative pathway dysfunction and lactate generation through excessive reliance on anaerobic metabolism to meet energy demands in PCC patients [32].

Glycine, an amino acid associated with neurotoxicity in excess [33–35], showed an increase in the COVID-19 group, persisting in the PCC group compared to the control group [269 (233–284) COVID group, 251 (234–266) Post-COVID group, 172 (127–224) µM control group, p<0.001]. In the opposite way, glutamate, the central nervous system's primary excitatory neurotransmitter, showed a decrease in PCC patients compared to both the COVID-19 group and the control group [93.8 (79.4–108) compared to 151 (122–205) and 137 (81.4–227) µM, p=0.007].

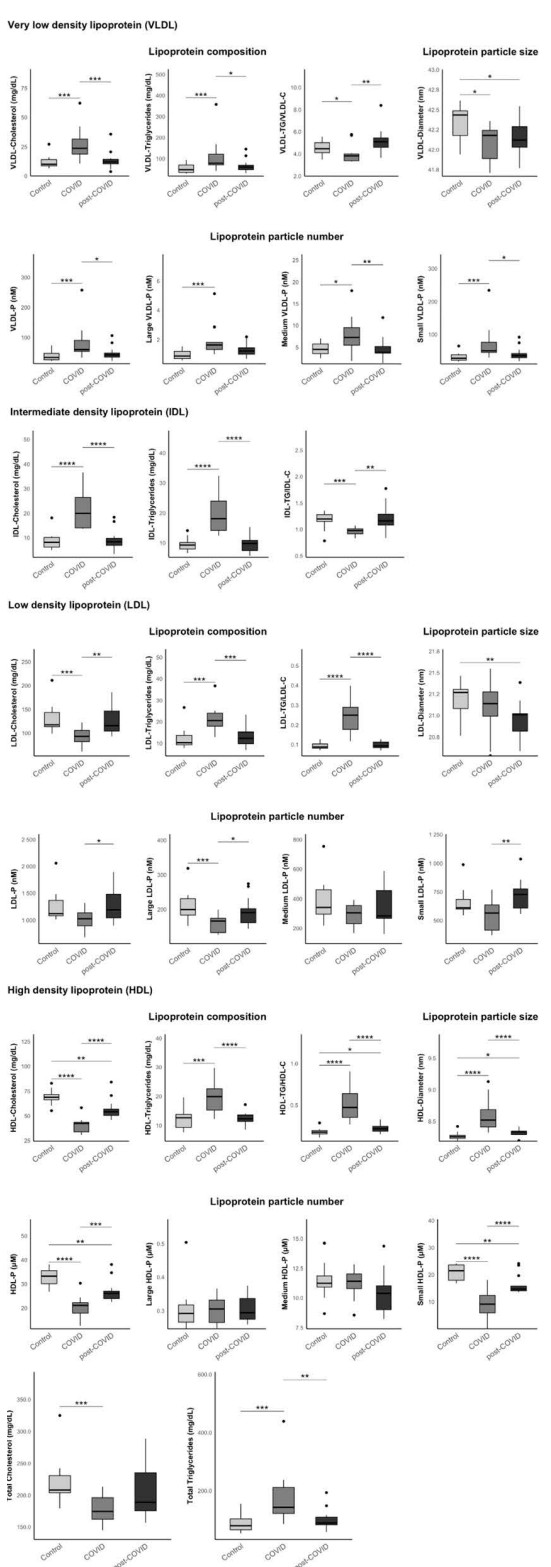

**Fig 1. Results of the lipoprotein analysis in the three study groups (control, COVID, and post-COVID) using 1H-NMR.** The data represent the median and interquartile range. Exploratory data analysis was performed using the Wilcoxon-U-Mann-Whitney test, adjusting the p-value with the Benjamini-Hochberg method for multiple comparisons.

**Glycoprotein Profile**

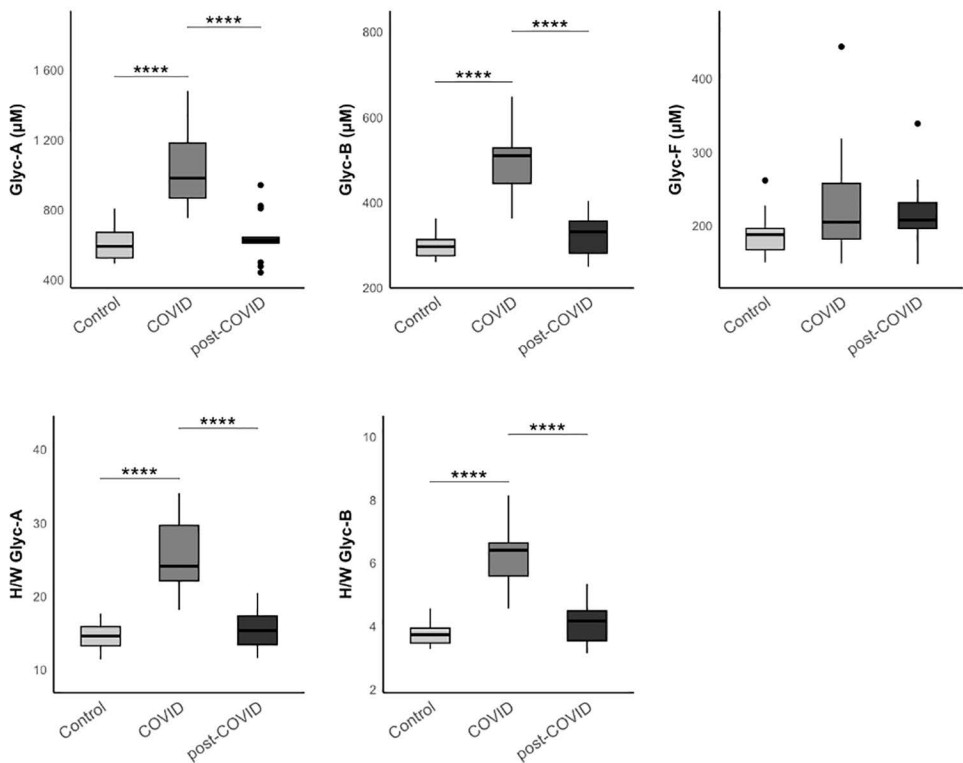

**Fig 2. Results of the glycoprotein analysis in the three study groups (control, COVID, and post-COVID) using 1H-NMR.** The data represent the median and interquartile range. Exploratory data analysis was performed using the Wilcoxon-U-Mann-Whitney test, adjusting the p-value with the Benjamini-Hochberg method for multiple comparisons.

Increased levels of glutamine, leucine, and isoleucine were observed in PCC patients after the acute infection phase compared to the control group: glutamine [369 (328–394) COVID group, 346 (299–391) Post-COVID group, 265 (173–281) µM control group, p = 0.001], leucine [145 (111–154) COVID group, 76.2 (73.1–84.4) Post-COVID group, 63.2 (61.9–69.5) µM control group, p < 0.001], isoleucine [57.0 (44.7–91.4) COVID group, 39.9 (29.4–42.3) Post-COVID group, 17.7 (17.1–23.2) µM control group, p < 0.001]. These amino acid increases indicate heightened protein catabolism. Results are shown in Fig 4. S4 Table.

## Discussion

Through physical performance tests (dynamometry, 6MWT, ergoespirometry), the true physical impact on these patients can be quantified. Data presented show, for example, reduced muscle strength as measured by dynamometry. Peripheral muscle involvement after acute infection is suspected, leading to physical deconditioning. The results of our tests seem to point in this direction, due to the poor physical performance described in the dynamometry, 6-minute walk test, and ergoespirometry. Previous studies have aimed to identify and to address this loss of physical capacity [36], while promoting physical exercise and rehabilitation programs as potential solutions to this condition. [11–14,37].

Lipidomic and metabolomic profiles of PCC patients compared to healthy controls and acute COVID-19 patients reveal persistent alterations potentially related to the symptoms experienced by these patients. During the acute infection phase, lipoprotein analysis indicated a pro-atherogenic environment: medium and small VLDL and IDL particles rich in

**Lipid Profile**

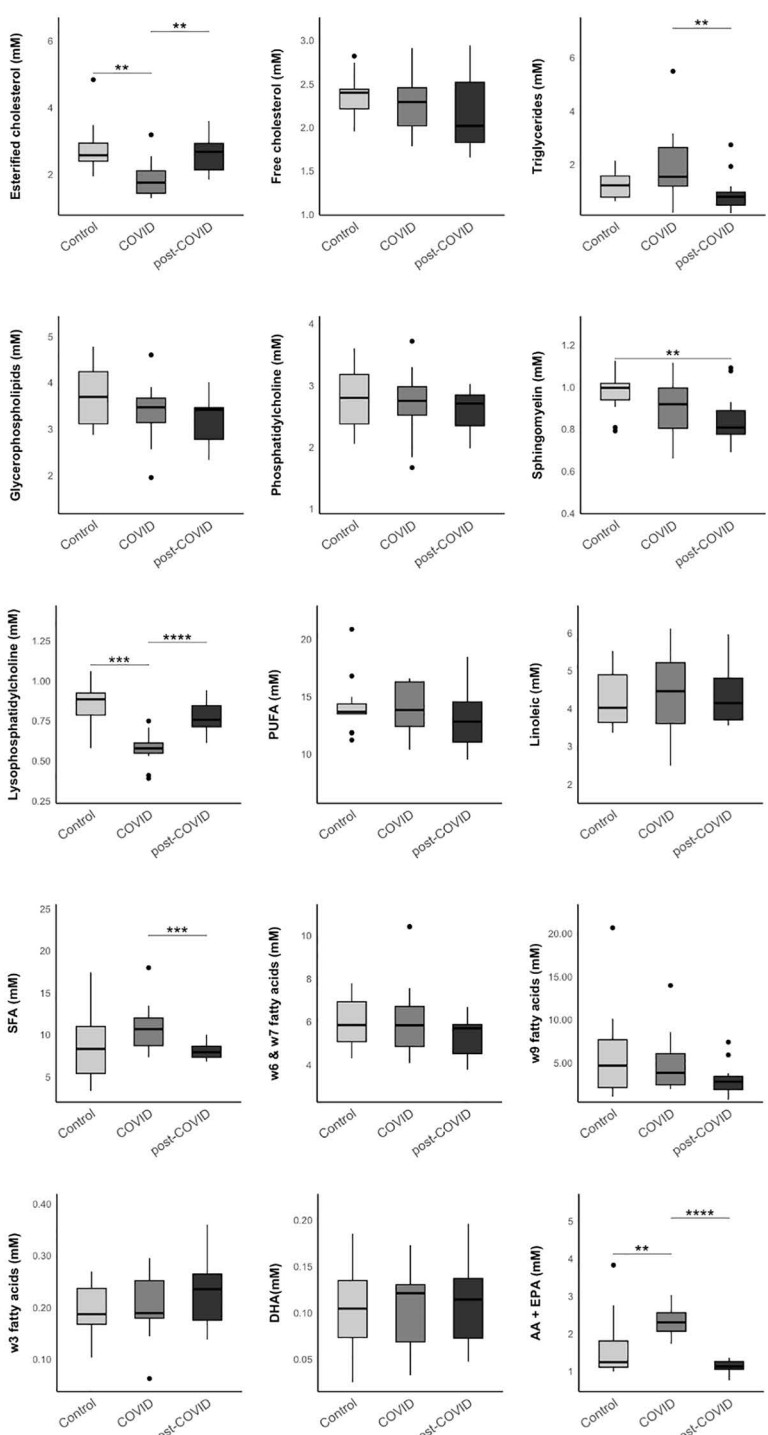

**Fig 3. Results of the lipid analysis in the three study groups (control, COVID, and post-COVID) using 1H-NMR.** The data represent the median and interquartile range. Exploratory data analysis was performed using the Wilcoxon-U-Mann-Whitney test, adjusting the p-value with the Benjamini-Hochberg method for multiple comparisons.

Low Molecular Weight Metabolite Profile

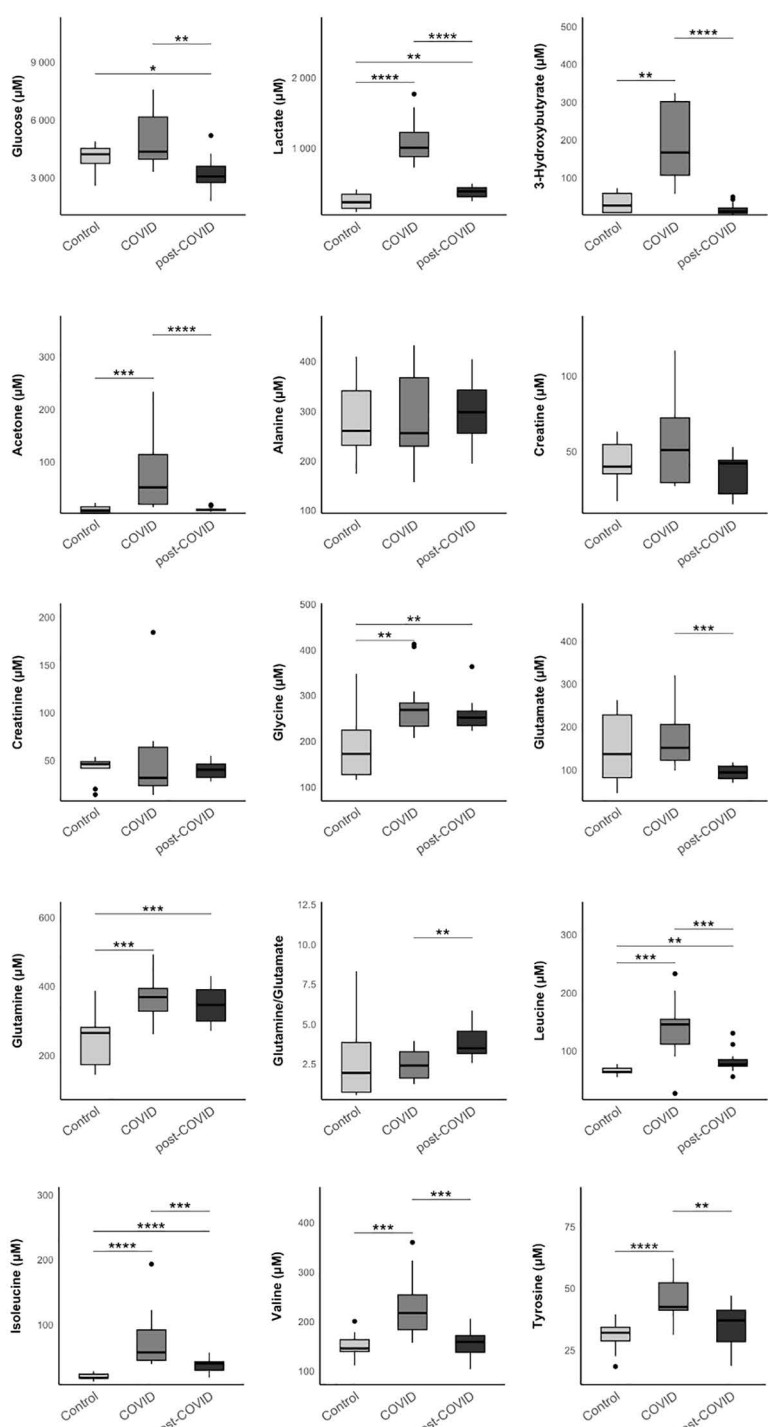

**Fig 4. Results of the low molecular weight metabolite (LMWM) analysis in the three study groups (control, COVID, and post-COVID) using 1H-NMR.** The data represent the median and interquartile range. Exploratory data analysis was performed using the Wilcoxon-U-Mann-Whitney test, adjusting the p-value with the Benjamini-Hochberg method for multiple comparisons.

triglycerides, as well as reduced HDL-cholesterol replaced by HDL-triglycerides, all of which promote atherogenesis. Such changes have been linked to disease severity and poor prognosis in multiple previous studies [38–41].

It is interesting to observe how some of these changes persist in PCC patients; such is the case with the decrease in HDL-cholesterol and medium and dense HDL particles, contributing to the maintenance of an atherogenic condition after the acute infection has resolved. Additionally, we must consider current evidence showing that HDL has a pleiotropic effect on autoimmune and inflammatory diseases, thanks to its potential anti-inflammatory capacity [42,43]. Thus, a decrease in HDL could promote an inflammatory state and, in turn, lead to an increased cardiovascular risk.

It has not been possible to demonstrate significant increases in glycoproteins in patients with PCC, which would have provided a good level of sensitivity for detecting underlying inflammation [18]. However, when focusing on the lipid profile, a statistically significant decrease in sphingomyelin was detected. Previous studies have linked alterations in the profile of sphingolipids and their metabolites with cardiovascular, renal, and metabolic diseases, due to their role in cell differentiation, proliferation, and apoptosis [44]. Similarly, alterations in sphingomyelin levels have also been associated with neurodegenerative diseases such as Parkinson or Alzheimer diseases [45–49].

Decreased glucose levels, combined with elevated lactate, could explain peripheral muscle fatigue and poor exercise tolerance in PCC patients, result from a tendency to rely on anaerobic mitochondrial energy pathways [32].

Continuing with low-molecular-weight metabolites, we want to highlight a potential alteration in neurotransmitters observed in patients with PCC. Specifically, an increase in glycine, an amino acid associated with neurotoxicity phenomena [33], such as hyperglycinemic encephalopathy in non-ketotic hyperglycinemia [34]. On the other hand, it is important to note that the decrease in glutamate levels identified in PCC patients has been linked to neuropsychiatric disorders like depressive disorder, supported by its role as the main excitatory neurotransmitter in the central nervous system [50,51].

Additionally, we found an increase in glutamine, leucine, and isoleucine levels in PCC patients, potentially indicating enhanced protein catabolism as a response to metabolic stress or increased metabolism of branched-chain amino acids (BCAAs), which is also associated with oxidative stress, inflammation, intense physical exercise, or muscle injuries [35].

However, the small sample size presents certain limitations when extrapolating the results obtained, with the possibility of less stable estimates or effects arising from chance. Nevertheless, the results presented and their correlation with clinical findings encourage the development of future studies with a more representative sample size.

## Conclusions

Significant changes in the metabolome and lipidome of PCC patients have been identified and highlight persistent alterations following the acute phase of COVID-19. Lipoprotein imbalances fostering atherogenesis and inflammation, disrupted aerobic energy metabolism with lactate overproduction and secondary muscle fatigue, and increased protein catabolism leading to oxidative stress and inflammation are key findings. These align with poor exercise tolerance noted in sports medicine analyses in these patients.

A larger study sample, as well as new analyses to demonstrate the evolution of these alterations over time in these patients, constitute some of the limitations of this work. Larger studies and further analysis over time are therefore necessary to elucidate the pathophysiology of this disease and better understand PCC's reality.

## Study data access plan

We will deposit all the primary research data from this study in the CSUC Research Data Repository (RDR, CORA RDR), a trusted, open, and FAIR-compliant repository managed by the "Consorci de Serveis Universitaris de Catalunya" (CSUC). This repository is based on the open-source Dataverse platform, assigns a persistent DOI to each dataset, and guarantees preservation of data for at least 10 years. Licenses under the Creative Commons framework (e.g., CC BY) are supported. Metadata will be curated and exposed via standard protocols reviewed by "Institut Investigació Sanitària Pere Virgili" (IISPV) dedicated platform, ensuring that data are findable and interoperable with major platforms such as OpenAIRE and EOSC.

For the personal data in this study, the datasets will be anonymized prior to publication in compliance with GDPR principles, applying recognized anonymization standards – such as removing direct identifiers and using techniques like hashing, noise-addition (data perturbation), or generalization.

## Supporting information

**S1 Fig. Representative 1H-NMR spectra from five aliquots of the same control serum sample, analyzed over five weeks with the LED sequence.**
(TIF)

**S2 Fig. 1H-NMR spectral processing and referencing.** Automated spectral processing is integrated into the Liposcale® software, ensuring identical conditions for all samples. Referencing is performed using the glycoprotein region (2.15–1.90 ppm), aligning the main peak at 2.034 ppm.
(TIF)

**S1 Table. Results of the lipoprotein analysis in the three study groups (control, COVID, and post-COVID) using 1H-NMR.** The variables are represented by the median and the interquartile range. The comparison between the different categories was performed using the Kruskal-Wallis test.
(DOCX)

**S2 Table. Results of the glycoprotein analysis in the three study groups (control, COVID, and post-COVID) using 1H-NMR.** The variables are represented by the median and the interquartile range. The comparison between the different categories was performed using the Kruskal-Wallis test.
(DOCX)

**S3 Table. Results of the lipid analysis in the three study groups (control, COVID, and post-COVID) using 1H-NMR.** The variables are represented by the median and the interquartile range. The comparison between the different categories was performed using the Kruskal-Wallis test.
(DOCX)

**S4 Table. Results of the low molecular weight metabolite (LMWM) analysis in the three study groups (control, COVID, and post-COVID) using 1H-NMR.** The variables are represented by the median and the interquartile range. The comparison between the different categories was performed using the Kruskal-Wallis test.
(DOCX)

## Author contributions

**Conceptualization:** Raúl Pavón, Sandra Parra, Francisco Javier Rubio.

**Data curation:** Raúl Pavón, Sandra Parra, Nuria Amigó, Lydia Cabau, Neus Martínez-Micaelo.

**Formal analysis:** Raúl Pavón, Sandra Parra, Nuria Amigó, Lydia Cabau, Neus Martínez-Micaelo.

**Investigation:** Raúl Pavón, Sandra Parra, Francisco Javier Rubio, Mireia Feliu, Marta Ríos, Simona Iftimie, Conxita Rovira, Antoni Castro.

**Methodology:** Raúl Pavón, Sandra Parra, Francisco Javier Rubio, Nuria Amigó, Lydia Cabau, Neus Martínez-Micaelo.

**Resources:** Sandra Parra, Nuria Amigó, Lydia Cabau, Neus Martínez-Micaelo.

**Software:** Nuria Amigó, Lydia Cabau, Neus Martínez-Micaelo.

**Supervision:** Sandra Parra, Francisco Javier Rubio, Antoni Castro.

**Writing – original draft:** Raúl Pavón.

**Writing – review & editing:** Sandra Parra, Francisco Javier Rubio, Antoni Castro.

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
