## [Decision Letter · Decision Letter 0]

24 Sep 2025

Dear Dr. Parra,

Thank you for submitting your manuscript to PLOS ONE. After careful consideration, we feel that it has merit but does not fully meet PLOS ONE’s publication criteria as it currently stands. Therefore, we invite you to submit a revised version of the manuscript that addresses the points raised during the review process.

We look forward to receiving your revised manuscript.

Kind regards,

Anil Bhatia, Ph.D

Academic Editor

PLOS ONE

Journal Requirements:

Reviewers' comments:

Reviewer's Responses to Questions

**Comments to the Author**

1. Is the manuscript technically sound, and do the data support the conclusions?

Reviewer #1: Partly

Reviewer #2: Partly

Reviewer #3: Yes

2. Has the statistical analysis been performed appropriately and rigorously?

Reviewer #1: Yes

Reviewer #2: No

Reviewer #3: Yes

3. Have the authors made all data underlying the findings in their manuscript fully available?

Reviewer #1: No

Reviewer #2: No

Reviewer #3: Yes

4. Is the manuscript presented in an intelligible fashion and written in standard English?

Reviewer #1: Yes

Reviewer #2: Yes

Reviewer #3: Yes

Reviewer #1: The combination of physical performance testing with lipidomic profiling is definitely innovative. The manuscript may benefit from the following suggestions:

1. Describe the process/functions that determined the profile of glycoprotein. Add more references and justification.Discuss the limitations of the small sample size as well.

2. Authors should add information about NMR data acquisition and should report acquisition parameters, processing steps, and how metabolite assignments were validated. Raw spectra and representative examples in supplementary material would improve transparency. Mention pulse parameters, magnetic field strength, software for peak interpretation etc.

3. The findings are interesting but because of low sample size, they do seem a little speculative.

4. The authors have used random forests. Why has this been selected over other ML methods and were any other methods tried. What sort of variance importance measurement was applied, Gini, SHAP etc. With small sample sizes, RF can lead to overfitting.

5. While the statistical analysis is robust, it is informative to indicate the meaning of mathematical results and their implications. e.g. mean, SD, IQR. Some other informative plots can be incorporated as well (heat maps, PCA).

6. Reference 31 and 48 seem duplicate.

Reviewer #2: When were the 46 CCC patients diagnosed? When is the physical condition assessed? When were plasma samples obtained? The temporality of CCC diagnosis is crucial for the variables and samples obtained.

The authors did not report any results from the random forest analysis. Furthermore, the final model or evaluation or the most important variables are absent from figures or tables. Why are they described in the statistical analysis? What is the relevance of this if they are not reported?

In addition, the authors should report the R script used for the statistical analysis, specifically for the random forest analysis.

Reviewer #3: The authors have done a good work on the global health concern at certain extend about Post-COVID-19 Condition (PCC), or long COVID. A study involving functional assessments and NMR-based metabolomic and lipidomic profiling revealed notable impairments in muscle strength and exercise tolerance among PCC patients, primarily linked to peripheral muscle involvement. The findings highlight the potential role of systemic metabolic disruption in the prolonged symptomatology observed in PCC. The paper is a good fit for PlosOne and should be acceptable after some suggested revisions.

1. Are there any alternative analytical techniques to NMR that you have used to validate metabolic changes in PCC and control patients? It would be great to include one more validatory method to the metabolic changes.

2. How you manage the age-related differences and modulation to metabolic profile of post-COVID-19 conditions across different population groups? The major group you have studied were near 30-50year but is there any data available below 30year age group? It will be great if you can include that as well.

3. To what extent was the dietary intake or nutritional status of the study participants considered in the design and interpretation of the metabolic findings?

**Do you want your identity to be public for this peer review?** For information about this choice, including consent withdrawal, please see our Privacy Policy

Reviewer #1: No

Reviewer #2: No

Reviewer #3: **Yes:** Bhaval Parmar

---

## [Author Response · Author response to Decision Letter 1]

1 Dec 2025

Reviewer #1: The combination of physical performance testing with lipidomic profiling is definitely innovative. The manuscript may benefit from the following suggestions:

1. Describe the process/functions that determined the profile of glycoprotein. Add more references and justification. Discuss the limitations of the small sample size as well.

We thank the reviewers for this valuable comment. Regarding the sample size, it is indeed one of the limitations of this study. We have gladly added a more detailed discussion on the implications resulting from this sample size, as well as future challenges for its improvement. These additions are included in the Discussion section of the manuscript (page 18, lines 9–12). Likewise, we have gladly added a more detailed explanation of the results obtained in the glycoprotein profile (page 14, lines 8-14).

2. Authors should add information about NMR data acquisition and should report acquisition parameters, processing steps, and how metabolite assignments were validated. Raw spectra and representative examples in supplementary material would improve transparency. Mention pulse parameters, magnetic field strength, software for peak interpretation etc.

Following your recommendation, we have added detailed information regarding NMR data acquisition, processing parameters, and metabolite/lipoprotein signal validation (Pages 6, 7, 8 and 9). A dedicated section has also been incorporated into the Supplementary Material (S1 Fig. and S2 Fig.).

3. The findings are interesting but because of low sample size, they do seem a little speculative.

We appreciate the observation and take it into consideration as a reason to continue working on this project in order to obtain better and more representative results. Its limitation is discussed in the Conclusions section of the manuscript (page 18, lines 20-23), and a more detailed description has also been added in this revised version in the Discussion section (page 18, lines 9-12).

4. The authors have used random forests. Why has this been selected over other ML methods and were any other methods tried. What sort of variance importance measurement was applied, Gini, SHAP etc. With small sample sizes, RF can lead to overfitting.

We thank the reviewer for this observation. The reference to the Random Forest analysis was included in an earlier version of the manuscript but was not part of the final analysis presented in the paper. We agree that its mention could lead to confusion. Accordingly, we have removed this section and revised the Statistical Analysis paragraph to accurately reflect the analyses that were performed and reported in the Results section. No Random Forest analysis was finally included in this study.

The updated statistical analysis section now reads as follow: (page 9, lines 10-19).

Quantitative results were expressed as mean and standard deviation for continue variables with normal distribution or with the median and 25–75 interquartile range otherwise, while qualitative or dichotomous variables were presented as percentages. To compare proportions and assess relationships between qualitative variables, we used the Chi-square test (χ²) or Fisher’s exact test, depending on sample size and expected frequencies. For comparisons of quantitative variables between groups, ANOVA or Kruskal–Wallis tests were applied, and Wilcoxon tests were used for pairwise non-parametric comparisons. In addition, Spearman’s Rho correlation coefficients were calculated to estimate associations between continuous variables. All statistical analyses were performed using R statistical software (version 4.3.3, R Foundation for Statistical Computing, Vienna, Austria).

5. While the statistical analysis is robust, it is informative to indicate the meaning of mathematical results and their implications. e.g. mean, SD, IQR. Some other informative plots can be incorporated as well (heat maps, PCA).

We understand the observation. In this revised version of the manuscript, the meaning of the mathematical results is described in greater detail. Please note the update and correction of the statistical analysis section, along with the corresponding secondary changes.

6. Reference 31 and 48 seem duplicate.

We appreciate the observation. Indeed, that reference was duplicated. This has been corrected in the revised version of the manuscript. It should be noted that, given a more extensive description of the glycoprotein profile with new references on the matter, the new numbering of that reference is now 33.

Reviewer #2: When were the 46 CCC patients diagnosed? When is the physical condition assessed? When were plasma samples obtained? The temporality of CCC diagnosis is crucial for the variables and samples obtained.

We agree that the temporal description is of vital importance for the subsequent validation of the results obtained in this work. In this revised version, we have added the dates during which the diagnostic process and recruitment of patients with Post-COVID-19 Condition were carried out. Likewise, the dates of sample collection, analysis, and later review are included, as well as the dates of the sports medical evaluation (Page 5, lines 3-6).

The authors did not report any results from the random forest analysis. Furthermore, the final model or evaluation or the most important variables are absent from figures or tables. Why are they described in the statistical analysis? What is the relevance of this if they are not reported? In addition, the authors should report the R script used for the statistical analysis, specifically for the random forest analysis.

We thank the reviewer for this observation. The reference to the Random Forest analysis was included in an earlier version of the manuscript but was not part of the final analysis presented in the paper. We agree that its mention could lead to confusion. Accordingly, we have removed this section and revised the Statistical Analysis paragraph to accurately reflect the analyses that were performed and reported in the Results section. No Random Forest analysis was finally included in this study.

The updated statistical analysis section now reads as follow: (page 9, lines 10-19).

Quantitative results were expressed as mean and standard deviation for continue variables with normal distribution or with the median and 25–75 interquartile range otherwise, while qualitative or dichotomous variables were presented as percentages. To compare proportions and assess relationships between qualitative variables, we used the Chi-square test (χ²) or Fisher’s exact test, depending on sample size and expected frequencies. For comparisons of quantitative variables between groups, ANOVA or Kruskal–Wallis tests were applied, and Wilcoxon tests were used for pairwise non-parametric comparisons. In addition, Spearman’s Rho correlation coefficients were calculated to estimate associations between continuous variables. All statistical analyses were performed using R statistical software (version 4.3.3, R Foundation for Statistical Computing, Vienna, Austria).

Reviewer #3: The authors have done a good work on the global health concern at certain extend about Post-COVID-19 Condition (PCC), or long COVID. A study involving functional assessments and NMR-based metabolomic and lipidomic profiling revealed notable impairments in muscle strength and exercise tolerance among PCC patients, primarily linked to peripheral muscle involvement. The findings highlight the potential role of systemic metabolic disruption in the prolonged symptomatology observed in PCC. The paper is a good fit for PlosOne and should be acceptable after some suggested revisions.

1. Are there any alternative analytical techniques to NMR that you have used to validate metabolic changes in PCC and control patients? It would be great to include one more validatory method to the metabolic changes.

We appreciate the importance of employing complementary analytical techniques to validate metabolic changes. However, in the present study, we chose to focus on magnetic resonance spectroscopy due to its well-established utility in assessing metabolic alterations in vivo. At the time of data collection, resource constraints limited our ability to incorporate additional methods such as mass spectrometry for example. Nonetheless, we fully recognize the value of multi-technique validation, and we consider the inclusion of alternative metabolic analyses as a relevant objective for future work.

2. How you manage the age-related differences and modulation to metabolic profile of post-COVID-19 conditions across different population groups? The major group you have studied were near 30-50year but is there any data available below 30year age group? It will be great if you can include that as well.

Indeed, this is a relevant observation. Age can play a role in modulating the metabolic profile of the patients studied. Since the objective of this study is precisely to compare and identify differences in the lipidomic and metabolomic profiles among patients with Post-COVID-19 Condition, patients with acute COVID-19 infection, and healthy controls, having a relatively homogeneous population in terms of age was considered useful to ensure a more representative comparison. To ensure that age is not a confounding factor, we matched the three groups by age. Furthermore, it is important to note that a large percentage of the patients diagnosed with Post-COVID-19 Condition in our hospital belong to the age range mentioned, with the diagnosis of patients either above or below this range being much less frequent. Also, we have detected that this group of patients are probably more impacted because of the worseness of physical condition in their work and ordinary live activities.

3. To what extent was the dietary intake or nutritional status of the study participants considered in the design and interpretation of the metabolic findings?

Once again, this is an excellent observation. Regarding nutritional status and its implications for the metabolome and lipidome, we observed a tendency toward overweight in patients diagnosed with Post-COVID-19 Condition, which could indeed influence the study results. To enable a better comparison and more reliable findings, it can be seen that the control and COVID-19 groups in the study present similar body mass index values. We can only assert that the patients included in this study were not under any kind of specific nutritional supplements. Likewise, data on hypertension and diabetes mellitus are included, since these could also be related to the metabolic condition of the patients involved. However, we agree that for a more thorough assessment, having a nutritional diary for the included patients would improve the quality of the work. This represents a challenge for future studies on our part, as would the inclusion of additional data such as patient ethnicity, due to its known influence on metabolism.

---

## [Decision Letter · Decision Letter 1]

4 Jan 2026

Assessment of Physical Status and Analysis of Lipidomic and Metabolomic Alterations in Patients with Post-COVID-19 Condition

PONE-D-25-39147R1

Dear Dr. Parra,

We’re pleased to inform you that your manuscript has been judged scientifically suitable for publication and will be formally accepted for publication once it meets all outstanding technical requirements.

Kind regards,

Anil Bhatia, Ph.D

Academic Editor

PLOS One

Additional Editor Comments (optional):

Reviewers' comments:

Reviewer's Responses to Questions

**Comments to the Author**

Reviewer #4: All comments have been addressed

Reviewer #5: All comments have been addressed

2. Is the manuscript technically sound, and do the data support the conclusions?

Reviewer #4: Partly

Reviewer #5: Yes

3. Has the statistical analysis been performed appropriately and rigorously?

Reviewer #4: Yes

Reviewer #5: Yes

4. Have the authors made all data underlying the findings in their manuscript fully available?

Reviewer #4: Yes

Reviewer #5: Yes

5. Is the manuscript presented in an intelligible fashion and written in standard English?

Reviewer #4: Yes

Reviewer #5: Yes

Reviewer #4: The study is unique in the sense that it combines exercise testing with blood profiling by NMR, which is a useful and an innovative pairing to understand PCC symptoms. While it sets up a solid groundwork for future research, it remains short in many aspects listed below and can be improved upon in consequent studies.

1. The study presented cross‑sectional snapshots where each group was measured once. This can show differences but it can’t show whether PCC changes persist over time, or whether they were present in the individuals before COVID. These factors are critical to biomarker determination.

2. The “acute COVID” group all had ICU‑level severe illness. PCC usually follows a wide range of illness severities. Comparing PCC to critically ill ICU patients can exaggerate differences that are really about illness severity, treatments, or hospitalization.

3. In addition to the nutritional diary suggested by another reviewer, few more data points such as other medications (e.g., steroids), fasting status, time of day of blood draw, and time since infection could also change blood lipids and amino acids a lot and may even explain some of the differences seen. This factors should be taken into account for normalization of "Background data".

4. Analyses done by the authors are mostly simple group-to-group comparisons. Adjustment needs to be done for important factors (age, sex, BMI, diabetes, medications) in the relationships between blood markers and fitness. That limits how confidently we can link the chemistry to the symptoms.

5. The timing of blood sampling and exercise tests after infection need to be consistent. Without consistent timing, differences may reflect when people were measured rather than actual PCC itself.

6. Since fasting and pre‑test conditions aren’t stated, it is hard to say decreased glucose and elevated lactate reflects “mitochondrial problems.” [Page 17, paragraph 4].

7. GlycA was high only in acute COVID, not in PCC. So, linking low HDL in PCC to active inflammation is only speculative without other inflammatory markers [Page 17, paragraph 2]. Other inflammatory biomarkers need to be profiled to prove the correlation.

8. Table 1 lists results that demonstrate poor physical condition. However, more details need to be included to show why the exercise tolerance is low. "Peripheral muscle" could be plausible but it seems like a leap without ruling out other causes that can cause this.

Conclusion - Despite the limitations, the manuscript feels like a good fit for PLOS ONE. It’s careful about what the data can and can’t say, the authors have responded thoughtfully to reviewer feedback, and the results are presented with the right amount of caution. As an exploratory, hypothesis‑generating study with clear methods and openly stated limitations, it sits comfortably within PLOS ONE’s scope.

Reviewer #5: The manuscript is well written and the responses to the reviewer comments are adequately and positively answered. Also, necessary changes based on reviewer's comments have been made leading to increase in the quality of the publication. However, there is one minute observation the authors should check. It is on Page 15 > Para: Analysis of low molecular weight metabolites >line 7 > p=0.005. Have the authors kept p-value 0.005 on purpose or it should be p<0.005.

**Do you want your identity to be public for this peer review?** For information about this choice, including consent withdrawal, please see our Privacy Policy

Reviewer #4: No

Reviewer #5: No

---

## [Editor Report · Acceptance letter]

PONE-D-25-39147R1

PLOS One

Dear Dr. Parra,

I'm pleased to inform you that your manuscript has been deemed suitable for publication in PLOS One. Congratulations! Your manuscript is now being handed over to our production team.

Kind regards,

on behalf of

Dr. Anil Bhatia

Academic Editor

PLOS One